# A Multitarget Approach to Evaluate the Efficacy of *Aquilaria* *sinensis* Flower Extract against Metabolic Syndrome

**DOI:** 10.3390/molecules27030629

**Published:** 2022-01-19

**Authors:** Hee-Sung Chae, Olivia Dale, Tahir Maqbool Mir, Bharathi Avula, Jianping Zhao, Ikhlas A. Khan, Shabana I. Khan

**Affiliations:** 1National Center for Natural Products Research, School of Pharmacy, The University of Mississippi, Oxford, MS 38677, USA; hchae@olemiss.edu (H.-S.C.); ordale@olemiss.edu (O.D.); tmmir@olemiss.edu (T.M.M.); bavula@olemiss.edu (B.A.); jianping@olemiss.edu (J.Z.); ikhan@olemiss.edu (I.A.K.); 2Department of Biomolecular Sciences, School of Pharmacy, The University of Mississippi, Oxford, MS 38677, USA

**Keywords:** *Aquilaria sinensis*, Thymelaeaceae, PPAR, LXR, adipogenesis, glucose uptake

## Abstract

*Aquilaria sinensis* (Lour.) Spreng is known for its resinous secretion (agarwood), often secreted in defense against injuries. We investigated the effects of *A. sinensis* flower extract (AF) on peroxisome proliferator-activated receptors alpha and gamma (PPARα and PPARγ), liver X receptor (LXR), glucose uptake, and lipid accumulation (adipogenesis). Activation of PPARα, PPARγ and LXR was determined in hepatic (HepG2) cells by reporter gene assays. Glucose uptake was determined in differentiated muscle (C2C12) cells using 2-NBDG (2-deoxy-2-[(7-nitro-2,1,3-benzoxadiazol-4-yl)amino]-D-glucose). Adipogenesis was determined in adipocytes (3T3-L1 cells) by Oil red O staining. At a concentration of 50 µg/mL, AF caused 12.2-fold activation of PPARα and 5.7-fold activation of PPARγ, while the activation of LXR was only 1.7-fold. AF inhibited (28%) the adipogenic effect induced by rosiglitazone in adipocytes and increased glucose uptake (32.8%) in muscle cells at 50 μg/mL. It was concluded that AF acted as a PPARα/γ dual agonist without the undesired effect of adipogenesis and exhibited the property of enhancing glucose uptake. This is the first report to reveal the PPARα/γ dual agonistic action and glucose uptake enhancing property of AF along with its antiadipogenic effect, indicating its potential in ameliorating the symptoms of metabolic syndrome.

## 1. Introduction

Metabolic syndrome is defined by the World Health Organization (WHO) as a pathological condition characterized by hyperlipidemia, hypertension, abdominal obesity, and insulin resistance [1]. The incidence of metabolic syndrome often parallels the incidence of obesity and of type 2 diabetes [2]. Considering the association of metabolic syndrome with obesity, it is quite likely that in a high percentage of these predominantly obese patients, the dysfunction of their adipose tissue becomes a main contributor to the subsequent associated complications. In numerous studies, phytochemicals belonging to polyphenols and flavonoids have been found to improve metabolic syndrome through their beneficial effects against lipid accumulation, high blood pressure and increased blood glucose [3,4]. A number of medicinal plants have also been reported to exhibit promising effects against metabolic syndrome due to their antidiabetic, hypolipidemic and anti-obesity effects [5,6,7,8,9]. Peroxisome proliferator-activated receptors (PPARs) are nuclear receptors that belong to the family of ligand-inducible transcription factors, mainly expressed in three isoforms (PPAR α, β/δ and γ). They play important roles in maintaining the homeostasis of the adipose tissue through the regulation of glucose and lipid metabolism by modulating the expression of genes involved in several biological processes such as lipid and glucose metabolism, adipogenesis, inflammation, and energy homeostasis [10,11,12]. The function of PPARs is similar to steroid receptors, which can be activated by nutrients, nutraceuticals, dietary fatty acids and their metabolites and phytochemicals [13], thereby redirecting metabolism by acting as lipid sensors [14]. Hence, PPARs represent interesting pharmacological targets for the regulation of metabolic pathways such as insulin sensitivity, inflammatory signaling, fatty acid storage, lipid and glucose metabolism in the body, thereby modulating several of the underlying pathologies related to metabolic diseases [15]. Liver X receptor (LXR) is a member of the nuclear receptor family that is highly expressed in liver. LXR plays a key role in metabolic pathways by regulating various target genes involved in both lipid and glucose metabolism in liver [16]. Thus, LXR is also considered as a potential target for therapeutic intervention in diseases associated with dysregulated metabolism, including atherosclerosis, and type 2 diabetes [17].

*Aquilaria sinensis* belongs to the genus *Aquilaria* from family Thymelaeaceae [18,19,20]. The plant is native to China and is widely found in the southern provinces of mainland China (Guangdong, Guangxi, Hainan and Fujian) and Taiwan [18]. This plant is known for its resinous secretion (agarwood), often secreted by it in defense against injuries. In traditional Chinese medicine (TCM) and Ayurvedic medicine, agarwood from *Aquilaria* spp. has been used for a long time as a sedative, analgesic and antiemetic agent. It has commonly been used for the treatment of joint pain, inflammatory ailments, and diarrhea [20,21,22]. However, traditional uses have also been scientifically proven at the pre-clinical level, such as sedative, gastric and cardioprotective effects. *Aquilaria* spp. (agarwood) have been reported to be a rich source of bioactive secondary metabolites (sesquiterpenes, chromones, flavonoids, benzophenones, diterpenoids, triterpenoids and lignans), which exhibit a wide range of pharmacological effects [20,22,23]. Ethanol, methanol and water extracts of the Aquilaria leaves showed a presence of flavonoids, tannins and lignans. In China, topical application of Aquilaria leaves is used in treating fractures and bruises. In Korea, agarwood is used in treating cough and asthma due to its sedative effects [20]. Thus far, numerous pharmacological activities have been attributed to aerial parts of *A. sinensis*. Aromatic dark resinous heartwood of *A. sinensis* is highly valuable in therapeutic perfumes, traditional medicine, religious purposes, and aromatic food ingredients [24]. The agarwood from *A. sinensis* has been reported for anti-inflammatory, analgesic, anti-depressive, and antiproliferative activities [25], while the leaves have also been reported for anti-inflammatory, anti-obesity and anti-diabetic properties [26,27]. The leaves and flowers of *A. sinensis* are widely consumed in the form of herbal tea in the southern part of China for various health benefits in alleviating symptoms of inflammation, anaphylaxis, fractures and bruising [21]. The chemical constituents of *A. sinensis* leaves and their biological activities have been described in detail earlier [20,22]. However, there is not much known about *A. sinensis* flowers except for few reports on their medicinal properties and our earlier report on the phytochemical analysis of flower buds of *A. sinensis* revealing benzophenone glycosides as the major components [28,29]. We also reported anti-oxidant and cytotoxic effects of the constituents isolated from the flower buds of *A. sinensis* [29]. Thus far, limited data are available on pharmacological activity of *A. sinensis* flowers. The present study was undertaken to investigate the effects of *A. sinensis* flowers on PPARα, PPARγ and LXR activation, adipogenesis and glucose uptake in hepatocytes, adipocytes and differentiated myoblasts with the aim to establish its potential in ameliorating the condition of metabolic syndrome. A qualitative analysis of the chemical constituents in *A. sinensis* flower extract was also carried out using ultra-high-performance liquid chromatography with quadrupole time-of-flight mass spectrometry (UHPLC-QToF-MS).

## 2. Results and Discussion

Maintenance of energy homeostasis and the response to environmental and nutritional conditions require the coordination of multiple organs and tissues such as liver, muscle and adipose tissue [30]. The coordination of metabolic regulation among organs and tissues, which requires communication among these organs and tissues, is essential for maintaining the homeostasis of metabolism, especially for glucose and lipid metabolism [31]. PPARs have multiple effects on liver, muscle, and adipose tissue [10]. PPARs are highly expressed in macrophages, adipose tissue and hepatocytes, and play important roles in adipogenesis, lipid metabolism, glucose homeostasis, and immune regulation [32].

PPARα agonists, such as ciprofibrate, are known for their lipid lowering (hypolipidemic) effects, whereas PPARγ agonists, such as rosiglitazone, are known for glucose lowering (hypoglycemic) effects and have been used in the treatment of hyperlipidemia and hyperglycemia. However, thiazolidinedione class of antidiabetic drugs (rosiglitazone, pioglitazone) acting through a full PPARγ agonistic effect, is hampered by adverse effects related to increased weight gain and fluid retention due to enhanced adipogenesis. It remains to be seen whether the dual PPARα/γ agonists have similar limitations [33]. Dual PPARα/γ agonists may combine the therapeutic effects of both drugs, with greater efficacy and advantage in the treatment of metabolic syndrome and type 2 diabetes, while exhibiting partial agonistic effects toward both PPARα and PPARγ. Luciferase reporter gene assays were employed to determine the agonistic activity of the methanolic extract of *A*. *sinensis* flowers (AF) toward PPARα and PPARγ in human hepatic cell lines (Figure 1). We found a dose dependent agonistic effect of AF on PPARα and PPARγ with 5.24-, 9.25- and 12.18-fold increase in PPARα (Figure 1A) and 2.48-, 3.66- and 5.71-fold increase in PPARγ (Figure 1B) at concentrations of 12.5, 25 and 50 μg/mL, respectively. Liver X receptor is known to be activated by endogenous oxysterols and to induce the target genes that correct sterol overload by promoting cholesterol disposal from the cell [34]. The identification of LXR as a regulator of cholesterol sensing and handling has led to sustained efforts to discover LXR agonists that might prove therapeutic against inflammatory diseases including atherosclerosis [35]. Our results show that increase in LXR activity did not exceed 1.75-fold in HepG2 cells upon treatment with 12.5, 25, or 50 μg/mL of AF (Figure 1C) and was not dose dependent. Hence, the LXR activation effect of AF was much less potent than the PPARα or PPARγ activation effect, indicating a selectivity of dual agonistic action toward PPARs. PPARs can complex with a number of co-activator proteins to regulate transcriptional activation of PPAR target genes. Binding of co-activators usually occurs at the AF-2 region, which has been located on helix 12 of the ligand-binding domain. AF might activate PPARα and PPARγ through AF-2 region on helix 12 of the ligand-binding domain [36]. Identifying natural products as potential antidiabetic agents with selective dual PPAR modulating effects without the adverse effects of a full PPARγ agonists (e.g., synthetic molecules) is highly desirable. Medicinal plants are known to be rich sources of ligands for nuclear receptors and have been considered as potential therapeutic agents that could work through multiple targets in treating cardiometabolic diseases [37].

One strategy for reducing the abnormalities associated with the utility of full PPARγ agonists is considering the use of partial agonists or dual PPARα and PPARγ agonists, which are known to improve both hyperglycemia and hyperlipidemia and reduce the risk of adipogenicity caused by full PPARγ agonists (such as thiazolidinediones) [37]. Adipocyte differentiation is controlled by post-transcriptional PPARγ regulation [38]. PPARγ full agonists induce adipogenesis and enhance differentiation of preadipocytes to mature adipocytes [39]. We have previously shown the antiadipogenic effects of selected medicinal plants (*T. nervosa, T. hirsuta* and *T. chebula*) and phytochemicals (gallotannins) that possess the property of dual PPAR agonists [7,40,41]. Due to the observed dual agonistic effects of AF, we further evaluated its effect on adipogenesis and lipid accumulation in mature adipocytes. However, AF did not show adipogenic effects, unlike rosiglitazone (data not shown). Additionally, incubation of preadipocytes (3T3-L1 cells) with 10 μM rosiglitazone resulted in a significant increase in lipid content of differentiated adipocytes, as shown in Figure 2. By contrast, AF at concentrations of 50 µg/mL showed a 28.09% decrease in rosiglitazone-induced adipogenesis. These results suggest that AF can antagonize the undesired side effect of a full PPARγ agonist and may prevent adipogenesis and the accumulation of cytoplasmic lipid droplets during the differentiation in 3T3-L1 cells. These findings are similar to dual agonists previously reported by us and others [7,40,41,42] and suggest that AF may have the potential for its use against obesity.

Since we have observed a significant induction of PPARγ by *A. sinensis* extract and PPARγ agonists such as rosiglitazone are reported to enhance glucose uptake by increasing insulin sensitivity, we evaluated its effect on glucose uptake. We conducted glucose uptake assay in differentiated myoblasts (C2C12 cells). As shown in Figure 3A,B, it is evident that *A. sinensis* extract at 50 µg/mL showed a 32.8% increase in glucose uptake as compared to the vehicle control, whereas the positive control, rosiglitazone (10 µM), showed a 36.9% increase in glucose uptake under similar experimental conditions (Figure 3B). However, at concentrations of 20 µg/mL of AF, the increase in glucose uptake was not as significant. Glucose transporter protein type4 (GLUT4) is a key component in glucose homeostasis and the removal of glucose from circulation [43]. Glucose uptake in skeletal muscle cells is dependent on the presence of GLUT4 on the surface membrane [44]. Further research is needed to elucidate the mechanism through which an increase in glucose uptake takes place in differentiated myoblasts upon treatment with AF. As mentioned above, AF equivalently modulated transcriptional activity of PPARα and PPARγ, which was comparable to the effects of well-known agonists such as ciprofibrate and rosiglitazone (Figure 1) and enhanced glucose uptake in skeletal muscle cells (Figure 3). These properties of AF resemble dual PPAR agonists reported earlier by us, such as *T. hirsuta, T. chebula* and *T. nervosa* and gallotannins, showing glucose uptake enhancing effects [7,40,41].

To further obtain an insight into the constituents of AF, phytochemical analysis of AF was carried out using ultra-high-performance liquid chromatography with quadrupole time-of-flight mass spectrometry (UHPLC-QToF-MS). Utilizing the high chromatographic resolution and separation capabilities of UHPLC with QToF-MS provides the structural characterization from accurate mass measurement for both MS and MS–MS experiments. The technique offers a significant advantage for rapid screening of compounds in complex matrices. In this study, non-targeted analyses were performed. Most of the compounds were detected in both negative and positive ion modes. In (-)-ESI-MS, the mass spectra of the chromatographic peaks showed deprotonated molecules [M − H]^−^ and also involved the use of the protonated molecules [M + H]^+^ ions in the positive ion mode. Twenty-seven compounds were tentatively identified based on the accurate mass spectra, fragment ions and literature [20]. The main classes of compounds identified in the methanolic extracts of *A. sinensis* flowers were organic acids (malic, succinic), phenolic acid (procatechuic acid), benzophenone iriflophenone glycosides (iriflophenone C-glucoside, iriflophenone di-C-β-D-glucopyranoside, iriflophenone-rhamnoside, iriflophenone-acetyl-rhamnopyranoside, aqulaside A-C), flavonoids (kaempferol glycosides, isorhamnetin glycoside, genkwanin) and xanthones (mangiferin, isomangiferin) (Figure 4, Appendix A, Appendix A). We screened the isolated compounds belonging to the benzophenone class, reported previously by us as major compounds of AF [29], for PPARα, PPARγ, and LXR activation effects, but they were found to be inactive (data not shown). However, a number of known phenolic compounds were identified in AF. Among the phenolic constituents of AF mentioned above, mangiferin, genkwanin and kaempferol have already been reported earlier as PPAR agonists [45,46,47]. Flavonoids have been reported to bind well with PPARs [48,49]. Hence, the PPAR agonistic activity of AF seems to be a combined effect of its phenolic constituents, while benzophenones do not seem to contribute to this activity. However, earlier studies have reported α-glucosidase inhibitory effects of benzophenones, which could add to the antidiabetic potential of Aquilaria [23].

This is the first report to reveal the PPARα/γ dual agonistic action, glucose uptake enhancing property and antiadipogenic effect of *A. sinensis* flower, indicating its potential in preventing the symptoms of metabolic syndrome.

## 3. Materials and Methods

### 3.1. Chemicals and Reagent

Ciprofibrate, rosiglitazone, T0901317, 2-(N-(7-Nitrobenz-2-oxa-1,3-diazol-4-yl) Amino)-2-Deoxyglucose (2-NBDG) and Hoechst 33342 were purchased from Cayman chemical (Ann Arbor, MI, USA). Oil red O, 3-Isobutyl-1-methylxanthine (IBMX), insulin and dexamethasone were purchased from Sigma-Aldrich (St. Louis, MO, USA). Fetal bovine serum (FBS) was obtained from Hyclone Lab, Inc. (Logan, UT, USA). Phosphate-buffered saline (PBS) (pH 7.4), Hanks balanced salt solution (HBSS), minimal essential medium (MEM), Dulbecco’s modified Eagle medium (DMEM), DMEM/F12, trypsin EDTA, sodium pyruvate, HEPES, streptomycin, and penicillin-G were from GIBCO BRL (Invitrogen Corp., Grand Island, NY, USA).

### 3.2. Extract Preparation

Authentic material of the dried flower buds of *A. sinensis* (voucher specimen #18377, authenticated by Dr. Vijayasankar Raman at NCNPR, University of Mississippi) was powdered and extracted with methanol using ultrasonic extraction as described earlier [27]. The extract was completely dried to remove the solvent and was dissolved in DMSO (20 mg/mL) prior to bioassays.

### 3.3. Phytochemical Analysis by Ultra-High-Performance Liquid Chromatography/Quadrupole Time of Flight Mass Spectrometry (UHPLC/QToF-MS)

The liquid chromatographic system was an Agilent Series 1290, and separation was achieved on an Agilent Poroshell 120 EC-18 (2.1 × 150 mm, 2.7 µm) column. The mobile phase consisted of water (A) and acetonitrile (B), both containing 0.1 % formic acid at a flow rate of 0.21 mL/min, with gradient elution of 0 min, 5% B; 20 min, 25% B; 30 min, 55% B; 35 min, 100% B. Each run was followed by a 5 min wash with 100% B and an equilibration period of 5 min with 5% B. The column temperature was set at 40 °C. The mass spectrometric analysis was performed with a QToF-MS/MS (Model #G6530A, Agilent Technologies, Palo Alto, CA, USA) equipped with an ESI source with jet stream technology using the following parameters: drying gas (N_2_) flow rate, 11.0 L/min; drying gas temperature, 250 °C; nebulizer, 35 psig, sheath gas temperature, 400 °C; sheath gas flow, 10 L/min; capillary, 3500 V; skimmer, 65 V; Oct RF V, 750 V; fragmentor voltage, 50 V. The sample collision energy was set at 45 eV. All the operations, acquisition and analyses of data were controlled by Agilent MassHunter Acquisition Software Ver. A.05.00 and processed with MassHunter Qualitative Analysis software Ver. B.07.00. Each sample was analyzed in both positive and negative modes in the range of *m*/*z* = 50–1300.

### 3.4. Cell Culture

HepG2 (human hepatoma cell line), 3T3-L1 (mouse embryonic fibroblast cell line) and C2C12 (mouse skeletal muscle cell line) cells were obtained from ATCC (Manassas, VA, USA). HepG2 cells were cultured in DMEM/F12 supplemented with 10% FBS, 2.4 g/L sodium bicarbonate, 100 μg/mL streptomycin, and 100 U/mL penicillin-G. Preadipocytes (3T3-L1 cells) were cultured in DMEM containing 10% BCS, 100 units/mL penicillin and 100 μg/mL streptomycin. C2C12 cells were cultured in DMEM containing 10% FBS and 100 U/mL penicillin/streptomycin sulfate. The incubator was set at 37 °C, 5% CO_2_, and 95% relative humidity.

### 3.5. Reporter Gene Assays for PPARα, PPARγ and LXR Activation

Reporter gene assays for the determination of PPARα, PPARγ and LXR agonistic effects were carried out in HepG2 cells as described previously, with some modifications [40]. The cells were transfected with either pSG5–PPARα and PPRE X3-tk-luc or pCMV–rPPARγ and pPPREaP2-tk-luc or pCMX-hLXRα and LXRE-tk-luc plasmid DNA (25 μg of each/1.5 mL cell suspension) by electroporation at 160 V for single 70 ms pulse using BTX disposable cuvette with a Square electroporator T820 (BTX, Holliston, MA, USA). Transfected cells were plated at a density of 5 × 10^4^ cells/well in 96-well tissue culture plates and grown for 24 h. After 24 h, the cells were treated with AF (12.5, 25 and 50 μg/mL) or ciprofibrate (10 μM) or rosiglitazone (10 μM) or T0901317 (1 μM). After an incubation for 24 h with the test samples, the cells were lysed and the luciferase activity was measured using a luciferase assay system (Promega, Madison, WI, USA). Fold increase in luciferase activity of sample-treated cells was calculated in comparison to vehicle treated cells.

### 3.6. Adipocyte Differentiation Assay and Quantification of Lipid Content in 3T3-L1 Cells

For the differentiation assay, preadipocytes were seeded in 48-well plates at a density of 2 × 10^4^ cells/well and maintained until 2 days post-confluence. After 2 days of confluency, the medium was replaced with DMEM containing 10% FBS, 10 μg/mL insulin, 1 μM dexamethasone, IBMX and test samples at various concentrations (day 2). After 2 days, the medium was replaced with DMEM containing 10% FBS, 10 μg/mL insulin and test samples (12.5, 25 and 50 µg/mL) (day 4). The culture was then maintained in 10% FBS/DMEM for an additional 4 days (day 8). The extent of adipogenesis in differentiated adipocytes was monitored by microscopy and the quantitation of lipid accumulation was performed by Oil red O staining as described below [41].

### 3.7. Oil Red O Staining

To visualize the morphological changes, differentiated adipocytes were stained with Oil red O using the method of Weiszenstein et al. (2016) with some modification [50]. After 8 days of 3T3-L1 differentiation in presence of samples, medium was removed from the wells. Cells were washed twice with PBS and fixed with 10% formalin for 30 min. Fixed cells were stained with Oil red O solution (0.6% Oil red O in 60% isopropyl alcohol) for 1 h at room temperature and washed twice with distilled water. The images of stained cells were taken on a BioTek Cytation5 imager (Biotek Instruments, Winooski, VT, USA). The stained lipid droplets were captured, and the sum of droplet area was calculated using Gen 5 (Bio Tek, Winooski, VT, USA).

### 3.8. Muscle Cell Differentiation and Glucose Uptake Assay

For the glucose uptake assay, myoblasts (C2C12) were seeded in 24-well plates at a density of 1 × 10^5^ cells/mL and maintained until 3 days post-confluence. C2C12 cells were grown in DMEM containing 10% fetal bovine serum (FBS) at 37 °C in a 5% CO_2_ atmosphere. Once the cells reached 100% confluence, the cell culture medium was changed to DMEM supplemented with 2% horse serum, and the cells were incubated for 6 days. For the 2-NBDG assay, cells were treated with or without 2-NBDG (50 μg/mL) or with 2-NBDG and desired concentrations of AF or rosiglitazone (10 µM) for 24 h in glucose and serum-free DMEM. Subsequently, the cell culture medium was removed, and cells were washed with phosphate-buffered saline (PBS) twice. After adding PBS to each well, the fluorescence intensity was recorded at excitation and emission wavelengths of 485 and 535 nm, respectively, using a SpectraMax Microplate Reader equipped with SpectraMax SoftMax Pro 6.5 software (Molecular Devices, Sunnyvale, CA, USA). The concentration of 2-NBDG was calculated according to the standard curve of the fluorescence intensity/2-NBDG concentrations. At the end, PBS was removed, and cells were stained with Hoechst33342 (20 μg/mL) in 1 mL of PBS. Cells were washed with PBS twice. After 5 min of incubation at room temperature, images were acquired with the objective of 10 × using the image multi-mode reader of Cytation5 (Biotek Instruments, Winooski, VT, USA). Hoechst33342 was used as a nuclear counterstain.

### 3.9. Statistical Analysis

For multiple comparisons, one-way analysis of variance (ANOVA) was performed, followed by Dunnett’s *t* test. Data from experiments are presented as means ± standard error of the mean (SEM). The number of independent experiments analyzed is given in the figure captions.

## 4. Conclusions

In conclusion, the transcriptional activation of PPARα, PPARγ and LXR by AF was associated with an increase in glucose uptake and was not associated with increased adipogenesis. These findings suggest that AF, due to its strong dual agonistic action toward PPARα and PPARγ, along with the glucose uptake enhancing effect, might ameliorate hyperlipidemia and hyperglycemia without the adverse effect of weight gain and increased adipogenesis. Hence, the results of this study confer potential value to *A. sinensis* flowers in the treatment of metabolic syndrome and obesity. Further studies are warranted in appropriate animal models to determine its in vivo efficacy.

## Figures and Tables

**Figure 1 molecules-27-00629-f001:**
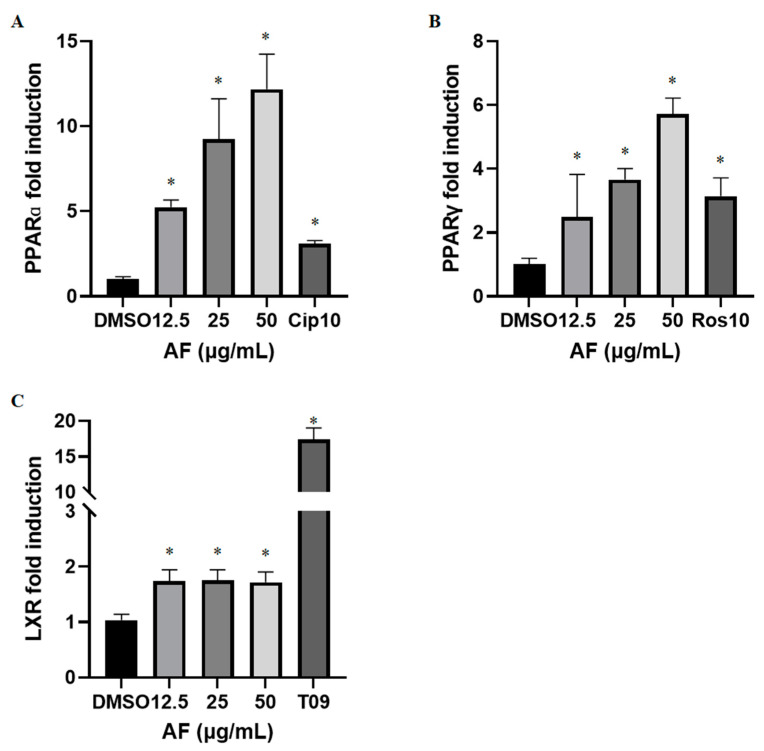
Increase in PPARα, PPARγ and LXR activity in HepG2 (human hepatocellular carcinoma) cells. Fold induction of PPARα (**A**), PPARγ (**B**) and LXR (**C**) was determined by luciferase assay in cells treated with AF (12.5, 25, and 50 μg/mL), ciprofibrate (CIP10, 10 µM), rosiglitazone (ROS10, 10 μM) and T0901317 (T09, 1 µM). The data are presented as mean ± SEM of duplicates in three independent experiments. * *p* < 0.05 compared to DMSO.

**Figure 2 molecules-27-00629-f002:**
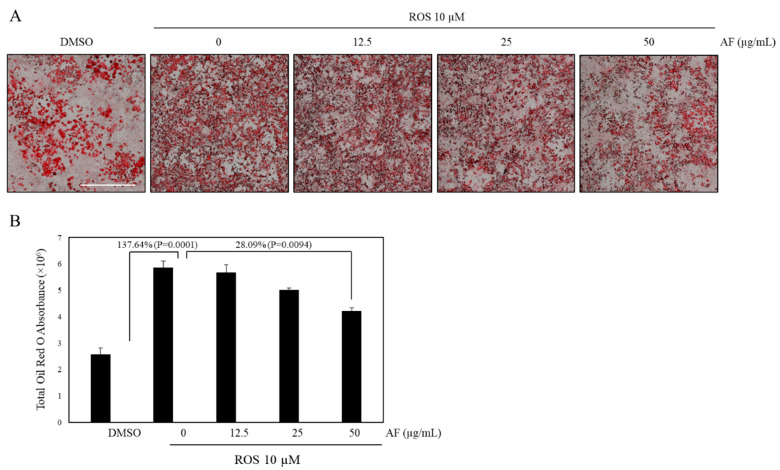
Suppression of rosiglitazone-induced adipogenesis. 3T3-L1 cells were exposed to 10% FBS/DMEM containing 10 μg/mL insulin, 1 μM dexamethasone, IBMX and 10 μM rosiglitazone in the presence or absence of AF at different concentrations (12.5, 25 and 50 µg/mL). (**A**) Effects of AF on rosiglitazone-induced adipocyte differentiation in 3T3-L1 cells (magnification: ×4; scale bar: 1 mm). (**B**) Quantitative analysis of adipocyte differentiation with rosiglitazone was assessed by spectrophotometric measurement of Oil-Red O-stained adipocytes. The data are presented as mean ± SEM of duplicates in two independent experiments. *p* values indicate pair comparison of DMSO vs. ROS (10 µM) and ROS (10 µM) vs. AF (50 µg/mL).

**Figure 3 molecules-27-00629-f003:**
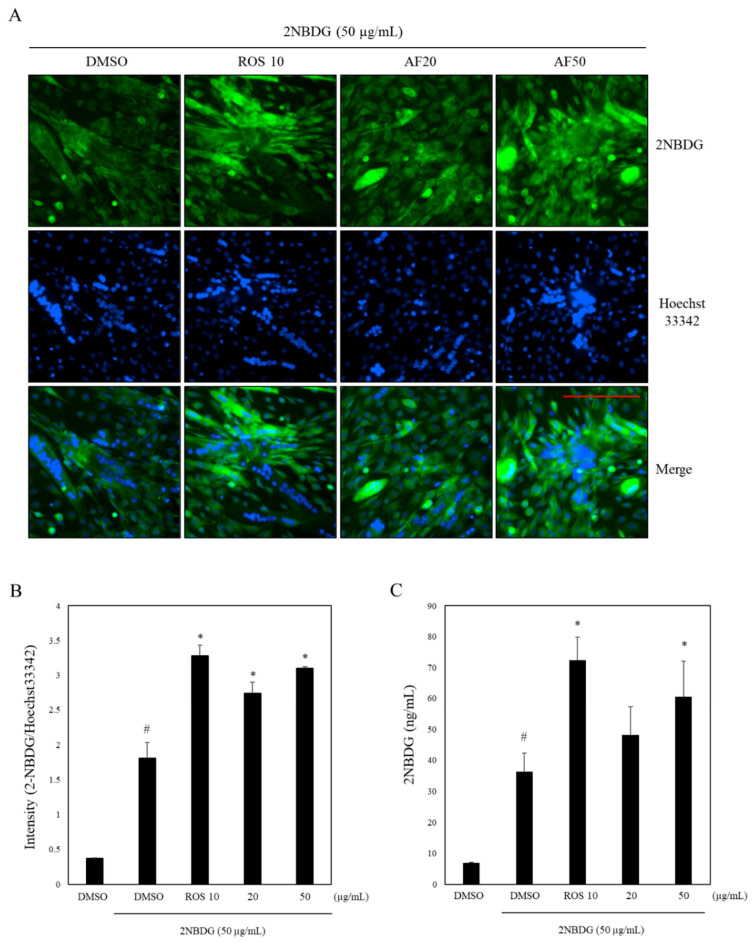
Increase in glucose uptake in differentiated myoblasts by AF. C2C12 cells were exposed to 2% horse serum for differentiation. After 6 days, cells were treated with AF (20 and 50 μg/mL) or rosiglitazone 10 μM (ROS 10). Glucose uptake was assessed by measuring 2-NBDG (50 μg/mL; ex/em ~475/550 nm) and Hoechst33342 (20 μg/mL; ex/em ~350/461 nm) uptake. (**A**) Representative images of cells untreated and treated with samples were acquired with 2-NBDG (in green), and Hoechst (in blue). Scale bar: 200 µm. (**B**) Relative intensity representing mean ± S.D. of triplicate in two independent experiments. (**C**) 2-NBDG uptake in ng/mL representing mean ± SEM. of triplicate in two independent experiments. # *p* < 0.05, compared with the DMSO treated group. * *p* < 0.05 compared with the 2-NBDG treated group.

**Figure 4 molecules-27-00629-f004:**
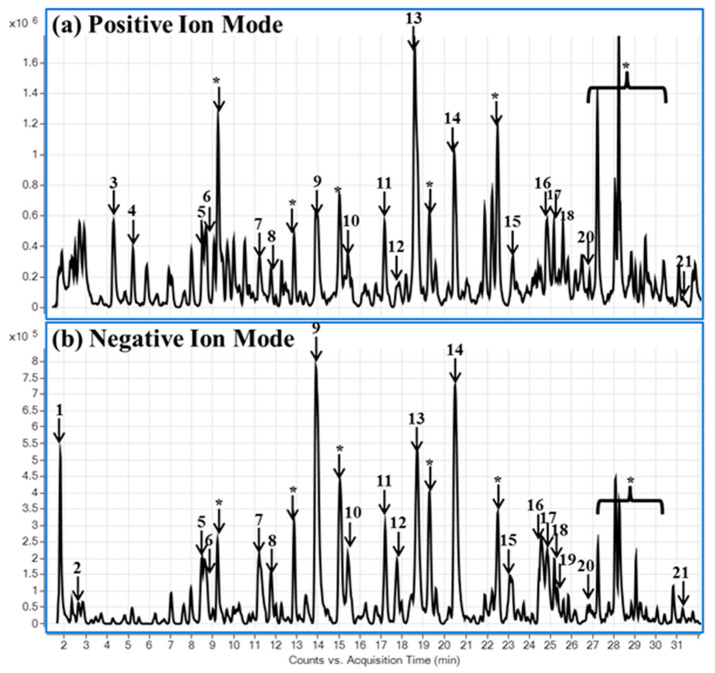
Phytochemical analysis of *A. sinensis* flower extract. LC-QToF base peak chromatograms are shown in (**a**) positive and (**b**) negative ion modes. The tentatively identified compounds are (1) malic acid; (2) succinic acid; (3) phenylalanine; (4) procatechuic acid; (5) iriflophenone 3-C-glucoside; (6) iriflophenone 3,5-di-C-β-D-glucopyranoside; (7–8) mangiferin/isomangiferin; (9) iriflophenone 2-rhamnoside; (10–13) iriflophenone 2-O-α-L-(4″-acetyl)-rhamnopyranoside/aquilarinenoside E; (14) kaempferol-3-O-β-D-glucopyranoside; (15) kaempferol diglucoside; (16) aquilarisinine; (17–18) kaempferol (*p*-coumaroyl-glucoside)/ kaempferol (*p*-coumaroyl-galactoside); (19) isorhamnetin coumarylglucoside; (20) kaempferol; (21) genkwanin; (*) unknown compounds.

## Data Availability

Not applicable.

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
