# Peer review of "A Multitarget Approach to Evaluate the Efficacy of Aquilaria sinensis Flower Extract against Metabolic Syndrome"

_molecules, 2022, doi:10.3390/molecules27030629_

Round 1

Reviewer 1 Report

The manuscript is interesting and its results have a good relevance.
However, I believe that the discussion needs to be further worked on. I would also like the authors to inform in the text perspectives on the potential use of this species and/or its metabolites worldwide and on its preclinical and clinical use in vivo.

Figs 2B and 3B, where is the statistical analysis?

Author Response

Reviewer #1

The manuscript is interesting and its results have a good relevance.
However, I believe that the discussion needs to be further worked on. I would also like the authors to inform in the text perspectives on the potential use of this species and/or its metabolites worldwide and on its preclinical and clinical use in vivo.

Response :We appreciate reviewer’s suggestion. Related information has been added in the ‘Introduction’ as suggested by the reviewer.

Figs 2B and 3B, where is the statistical analysis?

Response: Statistical analysis has been added in the revised version.

Reviewer 2 Report

The authors deal with the efficacy of Aquilaria sinensis flower extract on metabolic disorder. It has merit for publication, after some required revisions on the following points:

1) The authors assessed the extracts and reported the activity on PPAR α and γ. Since a difference is observed between the two receptors, any idea on the relationship-interactions between the constituents of the AF extracts and the receptor (interaction with active sites) will enhance the significance of the manuscript. Please consider this point and provide a small explanation/insight. 

2) I miss HRMS data. Though tentative identification is reported i do not see the monoisotopic masses, the MS/MS fragments. Yet, apart from the chromatograms at least 2-3 HRMS mass spectra should be provided (especially for bioactive components) and a detailed supplementary material. The latter will strengthen the manuscript. 

3) A comment on other organic solvents (ethanol) or combination with water for the extraction of flower is also welcome, since in phytochemical analysis different solvents can extract additional components (or different concentrations)

3) Minor comment, please use superscript in page 4, line 161 for [M-H]- 

Author Response

Reviewer #2

The authors deal with the efficacy of Aquilaria sinensis flower extract on metabolic disorder. It has merit for publication, after some required revisions on the following points:

1) The authors assessed the extracts and reported the activity on PPAR α and γ. Since a difference is observed between the two receptors, any idea on the relationship-interactions between the constituents of the AF extracts and the receptor (interaction with active sites) will enhance the significance of the manuscript. Please consider this point and provide a small explanation/insight. 

Response: A brief explanation about the interaction of constituents with the receptor has been added along with references in ‘Results and Discussion’ section.

2) I miss HRMS data. Though tentative identification is reported i do not see the monoisotopic masses, the MS/MS fragments. Yet, apart from the chromatograms at least 2-3 HRMS mass spectra should be provided (especially for bioactive components) and a detailed supplementary material. The latter will strengthen the manuscript. 

Response : We would like to thank for this suggestion that will certainly strengthen the paper. The suggested information is added as Supplementary material in the form of Table S1 and Figure S1.

3) A comment on other organic solvents (ethanol) or combination with water for the extraction of flower is also welcome, since in phytochemical analysis different solvents can extract additional components (or different concentrations)

Response : We have used methanol as our extraction solvent for this study and our phytochemical analysis was qualitative only. We agree that different solvents may result in different composition. But we also believe that the extraction efficiency of methanol is quite sufficient to extract majority of polar as well as some nonpolar components. The comparison of different solvents for extraction was not the focus of this study as we did not perform quantitative analysis. However, methanol, ethanol and water extracts of Aquilaria leaves were shown to have very similar phenolic content (please refer to Hashim et al 2016; Ref # 22).

3) Minor comment, please use superscript in page 4, line 161 for [M-H]- 

Response : Correction has been done on page 4.

Round 2

Reviewer 1 Report

The manuscript has been improved considerably. 
No further questioning.